# Theoretical and Practical Perspectives on what Influence Functions Do

**Andrea Schioppa**[1]  **Katja Filippova**[1]  **Ivan Titov**[2,3]  **Polina Zablotskaia**[1]
[1]Google DeepMind  [2]University of Edinburgh  [3]University of Amsterdam
{arischioppa, katjaf, polinaz}@google.com, ititov@inf.ed.ac.uk

## Abstract

Influence functions (IF) have been seen as a technique for explaining model predictions through the lens of the training data. Their utility is assumed to be in identifying training examples "responsible" for a prediction so that, for example, correcting a prediction is possible by intervening on those examples (removing or editing them) and retraining the model. However, recent empirical studies have shown that the existing methods of estimating IF predict the leave-one-out-and-retrain effect poorly. In order to understand the mismatch between the theoretical promise and the practical results, we analyse five assumptions made by IF methods which are problematic for modern-scale deep neural networks and which concern convexity, numeric stability, training trajectory and parameter divergence. This allows us to clarify what can be expected theoretically from IF. We show that while most assumptions can be addressed successfully, the parameter divergence poses a clear limitation on the predictive power of IF: influence fades over training time even with deterministic training. We illustrate this theoretical result with BERT and ResNet models. Another conclusion from the theoretical analysis is that IF are still useful for model debugging and correcting even though some of the assumptions made in prior work do not hold: using natural language processing and computer vision tasks, we verify that mis-predictions can be successfully corrected by taking only a few fine-tuning steps on influential examples.

## 1   Introduction and related work

Influence Functions (IF) [CS82] have been regarded as a tool that can trace model behavior on any example to the training examples [KL17, PLKS20, GRH+21]. Their theoretical justification lies in the ability to predict loss changes on a specific test point when training on a perturbed loss obtained by removing or down-sampling a given training point. In the case of an undesired model behavior on a test-point, the influential training examples for that test point have been assumed to be the ones "responsible" for the prediction so that intervening on those – e.g., by removing them and then *retraining* the model, or by taking additional fine-tuning steps [GRH+21] – would result in a change in the loss or prediction. It has been confirmed that for linear models it is indeed the case [KATL19].

Recently, in their extensive experiments, [BPF21] and [KS21] could not find empirical support for the claim that IF approximate the Leave-Some-Out Retraining (LSOR) effect on the loss in deep neural networks. In particular, they show that the correlation between the ranking of training examples produced by LSOR and the IF-based ranking is low and considerably affected by choice of hyperparameters. How can this discrepancy be explained? And, given that the theoretical justification for IF lacks empirical support, does it mean that IF should be abandoned as an explainability and debugging tool altogether?

In this work we clarify what question Influence Functions (IF) actually answer. We first identify assumptions which are either implicit or not investigated in prior work: these concern convexity,

numeric stability, training trajectory, and the the parameter divergence when retraining on a new loss. *In principle, any of these assumptions might be problematic and be the reason why the original theoretical justification for IF is not supported empirically.* However, we show how to address most of them successfully so that they cannot be the reason for the aforementioned discrepancy. Unfortunately, we also show that the parameter divergence is indeed problematic and requires to revise both theoretical and practical expectations about IF. This analysis paves the way to clarifying the question that IF can answer. As a first step in this direction, we need to distinguish two approaches to computing influence. The *Hessian-based Influence Functions (HIF)* [CS82, KL17] have a rigorous theoretical justification in statistics, but rely on strict convexity assumptions, which are not met in Deep Learning and which previous analyses of HIF in the Deep Learning literature still rely on [KL17, BNL$^+$22]. Our first contribution concerns HIF and is twofold: we prove (Theorem 1) that this unsatisfied convexity assumption is not as problematic for HIF as one may think–given a stationary point for the original loss, there is a nearby stationary point for the perturbed loss which can be approximated by using HIF. However, we point out a more serious problem with HIF: *there is no guarantee that by retraining on the perturbed loss one would get to that stationary point.* This observation provides an additional support to the second popular approach to IF, TracIn [PLKS20], which explicitly models the training dynamics and which additionally does not require to compute the expensive inverse Hessian vector product as it only uses gradient information.[1] Despite TracIn being grounded in the training dynamics, we unveil a hidden additive modeling assumption in [PLKS20] that *prevents it from correctly modeling the (re)-training dynamics.* Our second contribution is thus to provide a theoretical analysis (Theorem 2) of how training trajectories change when perturbing the loss. This is a key result as it suggests that two training trajectories differing by a small loss perturbation *could diverge over time to the point of violating a first-order expansion assumption that IF make*, thus making IF's predictions unreliable. Our further theoretical and empirical investigation confirms this conjecture. Therefore, *what IF can do is to predict parameter changes when fine-tuning for a limited number of steps on the perturbed loss. This requires to adjust how IF are evaluated (Section 5) and applied (Section 6).* In order to confirm the conjecture and re-adjust expectations regarding IF, we need a deeper analysis of the trajectory divergence.

Our third contribution is thus to validate the conjecture on trajectory divergence and its consequences for IF. We first prove (Theorem 3), using a discrete version of Gronwall's Lemma [Gro19], an upper bound on the parameter divergence when (re)-training on a perturbed loss. We then empirically verify that the bound is sharp in Section 5 and that IF can approximate parameter changes along the perturbed trajectory *only for a limited amount of time*. We then empirically verify that this leads to a *fading (of accuracy)* of IF predictions over time. Therefore, we theoretically demonstrate and empirically verify that IF can in general answer only what happens *when fine-tuning on a perturbed loss for a limited amount of time*, instead of a general retraining setting.

Therefore, our new theory indicates that IF have been used incorrectly as the emphasis has been on their LSOR potential. On the positive side, it suggests an alternative way of using IF that indeed yields substantial empirical improvement. To demonstrate that, as our final contribution, in Section 6 we propose and verify a very simple method for correcting mis-predictions by taking only a few gradient steps on influential examples. Our proposal is related to model editing [DCAT21, inter alia] in that the latter also aims at changing model predictions. However, there the model is given and the modifications are done on its parameters whereas IF aim at understanding how specific training examples are responsible for the current model behavior and editing predictions through the data. We leave for future work building connections between model editing and IF.

For the case of Hessian-based influence functions, the recent work of [BNL$^+$22] has proposed to resolve the discrepancy between the theory and the empirical evidence by evaluating them using the proximal Bregman response function. However, [BNL$^+$22] relies crucially on the convexity assumption: while under this assumption there is agreement between our work and their findings (e.g. our parameter divergence drives the linearization error [BNL$^+$22, Sec. 4.4]), our theory applies also the the case where the Hessian is not positive definite. Moreover, our work covers also the scope of Gradient-based influence functions including TracIn [PLKS20].

---

[1]To incorporate training dynamics, TracIn exploits multiple checkpoints which introduces a substantial overhead, hence only the latest checkpoint is often used in practice. Other gradient-based methods have been proposed in [CGFT19, HYHI20], but they lack a justification from the training dynamics perspective.

## 2 Definition of Influence Functions and Notation

The different methods proposed to define Influence Functions share a common goal: *forecasting the change in the prediction on a test example when up (or down-) weighting a training example*. This is achieved by tracing the effect that re-weighting a training example has on the model parameters.

Removing or adding a training point can be modeled by a *perturbation* of the loss function; such a perturbation can be made smooth by modeling the weighting of a training point by a continuous parameter $\varepsilon$. More generally, let $L(\theta)$ denote the loss function where $\theta \in \mathbf{R}^N$ are the model parameters. We model loss perturbations by introducing *a variation of the loss*, which is a smooth function $\mathcal{L}(\theta, \varepsilon)$ depending on an additional vector parameter $\varepsilon \in \mathbf{R}^Q$ and coinciding with the original loss for $\varepsilon = 0$. For example if $l_x$ denotes the loss on a given training point $x$, we set $\mathcal{L}(\theta, \varepsilon) = L(\theta) + \varepsilon l_x$. While the scalar case $Q = 1$ is the one commonly considered, e.g. [KL17, PLKS20], *we introduce the vector one $Q > 1$ which arises naturally when considering the effect of re-weighting multiple points differently*, e.g. when modifying the weights in a mixture of different data-sets.

The IF method of choice then predicts *what would happen if training on $\mathcal{L}(\theta, \varepsilon)$ instead of $L(\theta)$*. The final parameters are modeled as a function $\theta_\varepsilon$ of the perturbation parameter; *assuming that such a function is well-defined and sufficiently smooth*, one then makes a first order expansion $\theta_\varepsilon \simeq \theta_0 + \varepsilon^T \nabla_{\varepsilon|0}\theta_\varepsilon$, where $\nabla_{\varepsilon|0}\theta_\varepsilon$ is the $Q \times N$-dimensional Jacobian at $\varepsilon = 0$ (see also Section A).

Under this first order assumption it is then straightforward to measure the change in the loss $l_z$ corresponding to a test point $z$:

$$l_z(\theta_\varepsilon) - l_z(\theta_0) \simeq \varepsilon^T \nabla_{\varepsilon|0}\theta_\varepsilon \nabla_{\theta|\theta_0} l_z. \tag{1}$$

We emphasize that (1) is *general to different IF methods*, which *differ in the specific derivation* of $\nabla_{\varepsilon|0}\theta_\varepsilon$.

## 3 Problematic assumptions made by Influence Functions

### 3.1 Problematic Assumption #1: Convexity can be used to show that $\theta_\varepsilon$ is a function of $\varepsilon$

In order to show that for each value of $\varepsilon$ there is a single value of $\theta_\varepsilon$ (so that one can model the final parameters as a function of the perturbation parameter), HIF relies on strict convexity of $L$. While this assumption is realistic for the statistical models considered in [CS82], this is *not the case for neural networks*. Even when introducing a regularization term, the loss of a neural network is not even weakly convex, and optimization methods usually converge to saddle points [DPG+14]. To the best of our knowledge, previous analyses of HIF in the Machine Learning literature, e.g. [KL17, BPF21, BNL+22], *have relied on some form of strict convexity*. In Section 4.1 we will revisit HIF and prove (Theorem 1), roughly speaking, that near a given stationary point $\theta_0$ for the original loss $L$, there is a stationary point $\theta_\varepsilon$ of the perturbed loss $\mathcal{L}(\theta, \varepsilon)$ which can be modeled as a function of $\varepsilon$ and such that $\nabla_{\varepsilon|0}\theta_\varepsilon$ is given by $-H_{\theta_0}^{-1}\nabla^2_{(\varepsilon,\theta)|(0,\theta_0)}\mathcal{L}$ as in [CS82] (see Section A regarding the usage of $\nabla^2$).

### 3.2 Problematic Assumption #2: The model Hessian is not degenerate

As HIF requires to apply the inverse model Hessian to $\nabla^2_{(\varepsilon,\theta)|(0,\theta_0)}\mathcal{L}$, one needs to *ensure that inverting the Hessian is numerically stable*. It has been empirically demonstrated [GKX19] that most eigenvalues of the Hessian tend to cluster near $0$. Numerically, this results in a considerable source of errors and instabilities when estimating HIF; while regularization can alleviate this problem, it introduces a hyper-parameter in the definition of HIF; the minimal value of such a hyper-parameter ensuring numerical stability depends on the *smallest negative eigenvalue of the Hessian*. Unfortunately, in realistic settings, this can be *larger in absolute value than reasonable values for the regularization parameter*. For example in our ResNet experiments regularization is of the order $10^{-4}$, while the smallest negative eigenvalue is $\simeq -5$. In Section 4.2 we discuss how the Arnoldi-based Influence Functions (abbr. ABIF) (which were introduced by [SZTS22] for computational efficiency) can be used to address such instability issues.

### 3.3 Problematic Assumption #3: Training trajectory can be ignored in Hessian-based Influence

Even if we solve the Problematic Assumption #1 for HIF, there is no guarantee that when actually re-training from scratch on $\mathcal{L}(\theta, \varepsilon)$ one would converge to the $\theta_\varepsilon$ given by Theorem 1 because the training trajectory is disregarded in HIF. As optimization is performed via some form of stochastic gradient descent, [PLKS20] propose *TracIn* which averages gradient dot-products across checkpoints in order to take into account the path taken by the training process. Importantly, for a single checkpoint $\theta_0$ TracIn estimates $\nabla_\varepsilon \theta_\varepsilon$ as $-\nabla^2_{(\varepsilon,\theta)|(0,\theta_0)}\mathcal{L}$, so the inverse Hessian vector product does not need to be computed. While it seems that TracIn takes into account the training trajectory, in the next Assumption we identify an issue with the way it models the training trajectory.

### 3.4 Problematic Assumption #4: The training trajectory can be modelled additively

The analysis of TracIn in [PLKS20] is based on a first-order expansion of the final change of the loss of a test point in terms of the gradient steps across the training trajectory. While this argument seems mathematically convincing, it overlooks that if one point is removed, or slightly up-sampled / down-sampled, *the subsequent training trajectory is modified.* Let $\theta_{\varepsilon,t}$ denote the value of the parameters after $t$ steps when doing gradient descent on $\mathcal{L}(\theta, \varepsilon)$. Denoting by $T$ the end-time, TracIn derives

$$\nabla_{\varepsilon|0}\theta_{\varepsilon,T} = -\sum_{t=0}^{T-1} \eta_t \nabla^2_{(\varepsilon,\theta)|(0,\theta_{0,t})}\mathcal{L}, \tag{2}$$

where $\eta_t$ is the learning rate at time step $t$. Now, the right-hand side in formula (2) is purely additive in the time steps; and addition is commutative, so *the order of the time steps does not matter.* One way to see that this is problematic is by making $\mathcal{L}$ time-dependent so that it differs from $L$ only at a specific time step $t$. In this case $\nabla^2_{(\varepsilon,\theta)}\mathcal{L}$ would be non-zero only at $t$ and formula (2) would consist of a single term. However, we would expect the perturbation at $t$ *to affect the following time steps*, so we should have at least $T - t - 1$ terms on the RHS for (2). In Section 4.3 we compute $\nabla_\varepsilon \theta_{\varepsilon,T}$ looking at the whole training trajectory and *discover an additional first-order term that is missing from TracIn*: *this term models the dependency of a time step on the earlier ones.* Concurrent work [GWP+23] also criticizes the additive assumption in TracIn on empirical grounds and proposes to build (re)-training simulators which are unfortunately computationally expensive as a new simulator must be fitted on the training set for each test point.

### 3.5 Problematic Assumption #5: $\theta_\varepsilon$ can be expanded to first order in $\varepsilon$

After we derive a formula for $\theta_{\varepsilon,T}$ and $\nabla_\varepsilon \theta_{\varepsilon,T}$ we realize that the latter can grow in norm in $T$. However, if $\theta_{\varepsilon,T}$ can be Taylor-expanded in $\varepsilon$, we *need $\theta_{\varepsilon,T} - \theta_{0,T}$ to be $O(\varepsilon)$ and $\nabla_\varepsilon \theta_{\varepsilon,T}$ to be $O(1)$.* If this is not the case, the whole IF approach described in Section 2 breaks down because IF approximate the parameter change $\theta_{\varepsilon,T} - \theta_{0,T}$ using the Taylor expansion $\varepsilon^T \nabla_\varepsilon \theta_{\varepsilon,T}$, but *the conditions to apply such a Taylor expansion are not satisfied.*

In Section 4 we show how assumptions #1–#4 can be successfully addressed which makes them not as problematic as they may first appear. However, for assumption #5 we will see that it puts a substantial limitation on the predictive power of IF. At the same time it allows for a new, locally bound perspective on IF – that influence holds for a limited number of fine-tuning steps (Section 5). Based on this finding, in Section 6 we propose a simple approach to use IF to correct mis-predictions which is theoretically grounded and is in addition much less compute intensive than those that involve re-training (e.g. [KL17]).

## 4 Addressing the problematic assumptions

*Proofs of all results are in the Appendix and we provide some motivation for the proof in the main text.*

## 4.1 HIF does not need Assumption #1

Previous work [CS82, KL17, BPF21, BNL$^+$22] on Hessian-based Influence Functions (HIF) has assumed that $L$ *is strictly convex* in order to claim that 1) *the minimum is unique* so that $\theta_\varepsilon$ can be modeled as a function, and 2) to use the Implicit Function Theorem to differentiate through the optimality condition.

Here we *will just assume that the Hessian is not singular at $\theta_0$*; by requiring that the final gradients do not change as we change $\varepsilon$ we prove:

**Theorem 1.** *Assume that $\nabla \mathcal{L}$ is $C^k$ ($k \geq 1$) and let $\theta_0 \in \mathbf{R}^N$; assume that the Hessian $H_{\theta_0} = \nabla^2_{\theta|\theta_0} L$ is non-singular; then there exist neighborhoods $U$ of $\theta_0$ and $V$ of $0 \in \mathbf{R}^Q$, and a $C^k$-function $\Theta : V \to U$ such that $\Theta(0) = \theta_0$ and $\Theta(\varepsilon) \in U$ is the unique solution in $U$ of the equation*

$$\nabla_\theta \mathcal{L}(\Theta(\varepsilon), \varepsilon) = \nabla_\theta \mathcal{L}(\theta_0, 0). \tag{3}$$

*Moreover, the gradient of $\Theta$ at the origin is given by:*

$$\nabla_{\varepsilon|0} \Theta = -H_{\theta_0}^{-1} \nabla^2_{(\varepsilon,\theta)|(0,\theta_0)} \mathcal{L}. \tag{4}$$

*Idea of the proof.* Existence of $\Theta(\varepsilon)$ is formulated as a solution to the local problem eq. (3); building a local solution does not require convexity if one uses the Implicit Function Theorem. □

The requirement that $\nabla_\theta L(\Theta(\varepsilon), \varepsilon)$ is constant in $\varepsilon$ has allowed us to establish a link between the training under different losses $\{\theta \to \mathcal{L}(\theta, \varepsilon)\}_\varepsilon$. If we assume that $\theta_0$ is a *stationary point*, i.e. $\nabla_\theta L(\theta_0) = 0$, we can strengthen the conclusions:

**Corollary 1.** *Under the assumptions of Theorem 1:*

1. *If $\theta_0$ is a stationary point of $L$, then each $\Theta(\varepsilon)$ is a stationary point of the loss $\theta \mapsto \mathcal{L}(\theta, \varepsilon)$.*

2. *If $\theta_0$ is a local (strict) minimum of $L$, for $\varepsilon$ sufficiently small, $\Theta(\varepsilon)$ is a local (strict) minimum of the loss $\theta \mapsto \mathcal{L}(\theta, \varepsilon)$.*

3. *Let $Q = 1$ (hence epsilon is a scalar) with $\mathcal{L}(\theta, \varepsilon) = L(\theta) + \varepsilon l_x(\theta)$, where $l_x$ is the loss corresponding to a specific training point $x$. We then obtain the classical result [CS82]:*

$$\left. \frac{d\Theta}{d\varepsilon} \right|_{\varepsilon=0} = -(\nabla^2_\theta L(\theta_0))^{-1} \nabla_\theta l_x(\theta_0). \tag{5}$$

## 4.2 If Assumption #2 is not satisfied, use Arnoldi-based Influence Functions

Theorem 1 requires that $H_{\theta_0}$ is non-singular. If the Hessian is singular, *we just need to keep fixed those parameters that are responsible for the degeneracy*. More precisely, we diagonalize $H_{\theta_0}$; we let $P_1$ be the subspace spanned by the eigenvectors corresponding to the non-zero eigenvalues and let $P_0$ be its orthogonal complement, that is, the kernel of $H_{\theta_0}$. Up to an orthogonal transformation of the parameters, we can assume that $P_1$ is spanned by the first $N_1$-coordinates and decompose $\theta = (\vartheta, \varphi) \in \mathbf{R}^{N1} \times \mathbf{R}^{N2}$ so that $H_{\vartheta_0} = \nabla^2_\vartheta L((\vartheta_0, \varphi_0))$ is non-singular. We then apply Theorem 1 to the restricted variation $(\vartheta, \epsilon) \mapsto L((\vartheta, \varphi_0), \epsilon)$. In terms of the original parameters $\theta$, this means that the function $\Theta(\varepsilon)$ is constrained to lie in $\theta_0 + P_1$, keeping the $\varphi$-component constantly equal to $\varphi_0$. Concretely, we can approximate $P_1$ using the Arnoldi iteration; therefore, *we can address the failure of Assumption #2 by using Arnoldi-based Influence Functions* (ABIF) [SZTS22], which approximate $P_1$ using the subspace spanned by the eigenvectors corresponding to the top-k (in absolute value) eigenvalues of the Hessian. The fact that ABIF stabilizes the estimation of influence scores has been observed empirically in previous work: in [SZTS22, Fig. 2] where using ABIF instead of the full Hessian improves the retrieval of mislabeled examples, and in [FLP$^+$23, Fig. 3] where ABIF is the best solver on the QA task and is comparable to the other solvers on the text-completion task.

## 4.3 The training trajectory can be traced to address Assumptions #3–#4

We need to improve our notation to correctly trace the training trajectory. The first issue is to keep track of the parameters across the time steps; the second issue are *sources of non-determinism*,

e.g. batch selection or random state for dropout. When comparing training trajectories for different values of $\varepsilon$ we want our notation to account for sources of non-determinism, as they might increase the difference between the training trajectories.

To address the first issue, we let $\theta_{\varepsilon,t}$ be the value of the parameters after training on $\mathcal{L}(\theta, \varepsilon)$ for $t$ steps. In particular, $\theta_{\varepsilon,0}$ *denotes* the initial value condition that we assume held fixed at $\theta_{\mathrm{init}}$ for different values of $\varepsilon$. As random state is a function of the training step (e.g. the batch to use at step $t$), to address the second issue, we just need to allow both the loss and the variation to depend on the train step, denoting them by $L_t$ and $\mathcal{L}_t$.

To simplify the exposition and for consistency with [PLKS20] we assume that models are trained with stochastic gradient descent. Letting $\eta_t$ be the learning rate at step $t$ we prove:

**Theorem 2.** *Assume that the model is trained for $T$ time-steps with stochastic gradient descent. Denoting by $H_t$ the Hessian $\nabla^2_{\theta|\theta_{0,t}} L_t$, then the final parameters $\theta_{\varepsilon,T}$ satisfy:*

$$\nabla_{\varepsilon|0}\theta_{\varepsilon,T} = -\sum_{t=0}^{T-1} \eta_t \nabla^2_{(\varepsilon,\theta)|(0,\theta_{0,t})}\mathcal{L}_t - \sum_{t=0}^{T-1}\eta_t H_t \nabla_{\varepsilon|0}\theta_{\varepsilon,t}. \tag{6}$$

*Idea of the proof.* One can explicitly write a system of equations for $\theta_{\varepsilon,T}$ and apply the operator $\nabla_{\varepsilon|0}$ to the solution of this system. $\qquad\square$

Note that the second term on the RHS of (6), which is missing from (2), takes into account the *contribution of the earlier time steps* that is missing from the analysis of [PLKS20]. A practical consequence of this second term is that the norm of $\nabla_{\epsilon|0}\theta_{\varepsilon,T}$ might grow (in $T$) more quickly than (2) would suggest: this is closely related to Assumption #5 which requires $\nabla_{\varepsilon|0}\theta_{\varepsilon,T}$ to be $O(1)$. In particular, for a constant learning rate, while (2) suggests a linear growth in the time step $T$, we will empirically verify in Section 5.1 that the growth is super-linear.

In Section K we illustrate the additional modeling error introduced by TracIn when computing the Jacobian $\nabla_{\epsilon|0}\theta_{\varepsilon,T}$ in the case of retraining BERT with SGD.

### 4.4 Assumption #5 becomes problematic over time

To address Assumption #5 we need to look into the *parameter divergence* $\|\theta_{\varepsilon,T} - \theta_{0,T}\|$ between training on $L$ and $\mathcal{L}$. The training dynamics is a discrete version of an ODE for which uniqueness and differentiability of the solutions with respects to the initial conditions can be established by Gronwall's Lemma [Gro19]. The parameter $\varepsilon$ can itself be considered an initial condition and we can prove a discrete version of Gronwall's Lemma to bound the parameter divergence. For simplicity of notation we prove the result for stochastic gradient descent, but we also sketch in the Appendix how to modify the argument to deal with optimizers.

**Theorem 3.** *In the setting of Theorem 2 assume that, for $t \leq T$, $\theta_{\varepsilon,t}$ lies in a bounded region $R$ such that*

$$\sup_{t,\theta \in R} \|\nabla\mathcal{L}_t(\theta, \varepsilon) - \nabla\mathcal{L}(\theta, 0)\| \leq C\varepsilon \tag{7}$$

*and that for $\theta \in R$ each loss $\mathcal{L}_t(\theta, \varepsilon)$ and its gradient wrt. $\theta$ are $A$-Lipschitz in $\theta$. Then*

$$\|\theta_{\varepsilon,T} - \theta_{0,T}\| \leq C\varepsilon \sum_{s<T} \eta_s (1 + \exp(2A \sum_{s<T} \eta_s)). \tag{8}$$

*Idea of the proof.* If the time steps were infinitesimal, i.e. if time was made continuous, the evolution of $\theta_{\varepsilon,t}$ would be governed by an ODE; then the bound (8) would be straightforward by applying the classical Gronwall's Lemma [Gro19]. In our case we just need to modify the ODE arguments to work with discrete time. $\qquad\square$

Note that the bound (8) is quite pessimistic as it involves an *exponential of the integrated learning rate* $\sum_{s<T} \eta_s$. This means that as $\sum_{s<T} \eta_s$ increases, the parameter divergence is no longer $O(\varepsilon)$ and

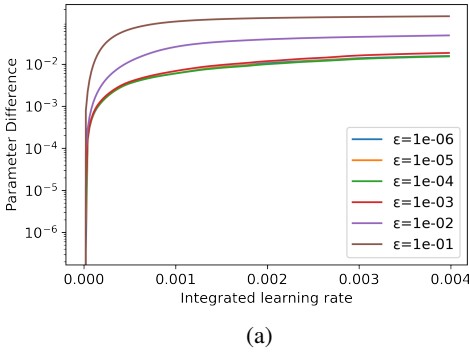 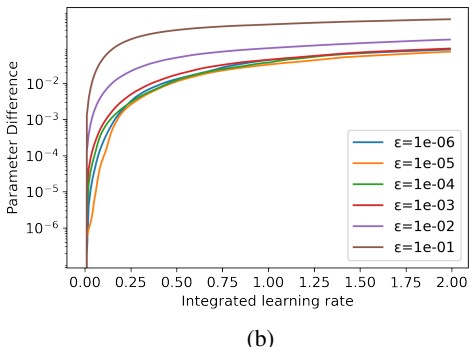

         (a)                                            (b)

Figure 1: Divergence of parameters (log-scale) as a function of the integrated learning rate. For each value of $\epsilon$ the divergence is exponential (corresponding to a line in log-scale) with two different divergence rates, one more steep at the beginning of (re)-training. (a) BERT, (b) ResNet

the crucial Assumption #5 is no longer satisfied. This observation leads to a few crucial conclusions:

1. An IF method can predict $\theta_{\varepsilon,t}$ only for a limited amount of time-steps: it is therefore incorrect to evaluate IF methods on LSOR or retraining from scratch.

2. IF methods need to be evaluated on what they can potentially do; therefore the evaluation setup should consist of fine-tuning on the perturbed loss only a limited amount of steps with evaluation metrics being reported as a function of the step.

3. Applying IF for correcting mis-predictions should also involve a time-bound scenario: we propose such a method in Section 6.

4. Sources of non-determinism between two training runs will likely increase the parameter divergence. So one should try to reduce this with *deterministic training*. For example, in the case of re-weighting a point $x$, i.e. setting $\mathcal{L} = L + \varepsilon l_x$, one should make sure to use the same batch $B_t$ for the loss $L$ at time step $t$ across training runs for different values of $\varepsilon$.

## 5 Illustrating the Theory

In this section we first demonstrate Theorem 3 empirically and then verify that the predictive power of influence scores degrades over time. Full details of our experimental setup are reported in the Appendix. We consider binary classification for nlp, where we fine-tune BERT on SST2; for computer vision we consider multi-class classification where we train from scratch ResNet on CIFAR10. *All our experiments use deterministic training*: the order of the training batches for the loss $L$ is held fixed across different runs, as well are the random generators when dropout is used.

### 5.1 Illustrating Parameter Divergence (Theorem 3)

Theorem 3 provides an upper bound when re-training on a perturbed loss. Such a bound is rather pessimistic as it involves the integrated learning rate. We therefore investigate empirically what happens with some typical Deep Learning setups. We take an intermediate checkpoint and keep training on a new loss obtained by up-sampling 16 training points with a weight $\varepsilon$, that is: $\mathcal{L}_t = L_{B_t} + \varepsilon \cdot L_B$, where $B_t$ is the training batch for step $t$ and $B$ is the batch of 16 points selected for up-sampling. For each time step $t$ we then compute $\|\theta_{\varepsilon,T} - \theta_{0,T}\|$ and then plot it against the integrated learning rate, see Figure 1.

Unfortunately, we observe that in these experiments *the upper bound in Theorem 3 is matched by a lower bound with the same exponential divergence*. We observe a first phase of quick divergence and then a second one in which the divergence is slower. For the second phase a linear fit of $\log \|\theta_{\varepsilon,T} - \theta_{0,T}\|$ against the integrated learning rate appears to be strong: for example for BERT we obtain an $R^2$ of at least $0.9$ across the different values of $\varepsilon$. The fitted slope, corresponding to $A$ in Theorem 3 depends on $\varepsilon$ and varies between 30 and 130.

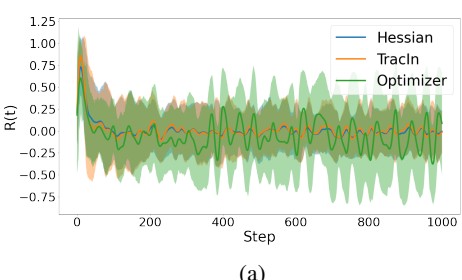 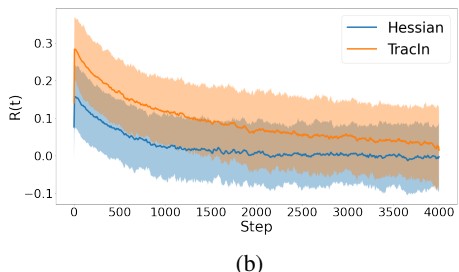

(a)                                    (b)

Figure 2: The predictive power of influence scores on the loss shifts degrades over time. (a) BERT, (b) ResNet. The line represents the average of the correlation $R(t)$ across runs, with the shaded area the corresponding 95% confidence region.

## 5.2 Illustrating the fading of influence

We now verify that the predictive power of influence scores fades over time. We again fix a model checkpoint $\theta_{\text{init}}$ and select 32 training points and 16 test points. For each training point $x$ we retrain on $\mathcal{L}_t^x = L_{B_t} - \frac{1}{100}l_x$, i.e. $x$ has been down-sampled; for each test point $z$ and time step $t$ we then compute the loss difference $\delta(z, x, t) = l_{z,x,t} - l_{z,t}$ where $l_{z,x,t}$ is obtained when (re-)training on $\mathcal{L}_t^x$ and $l_{z,t}$ is obtained when training on the vanilla loss $L_t = L_{B_t}$. Again, we have kept the order of the batches $B_t$ the same when re-training. Now, at the original checkpoint $\theta_{\text{init}}$ we can compute the influence scores $IF(z, x)$ for different methods, e.g. TracIn or HIF (using the the Conjugate Residual method[2]). For each time step we thus have $32 \times 16$ values of $\delta(z, x, t)$ that can be linearly regressed against $IF(z, x)$; *the corresponding Pearson correlation $R(t)$ then measures the predictive power of influence scores on the loss shifts when re-training*. We repeat the experiments for ResNet 9 times and for BERT 25 times, with a different selection of train and test points, so that we obtain confidence intervals for the resulting time-series $R(t)$. Ideally, the theory behind an influence method predicts $R(t) \sim 1$. However, as discussed above, Assumption #5 is indeed problematic and, because of the parameter divergence illustrated in 5.1 we expect $R(t)$ to degrade over time.

As Figure 2 demonstrates, this is indeed the case. The predictive power is high for BERT after a few re-training steps and then degrades quickly oscillating around 0 (which is covered by the confidence intervals). For ResNet, the predictive power is never as high, *but it still degrades monotonically over time, as predicted by the theory*. As we see in Section 6, the proponents retrieved for ResNet are less effective at correcting mis-predictions than those retrieved for BERT: we conjecture that this is related to the worse predictive power of IF in the case of ResNet. As BERT was trained with the Adam Optimizer, we also considered a variant which takes into account the optimizer's pre-conditioner by multiplying the gradients by the square root of the pre-conditioning matrix. In this case the predictive power is worse than for the vanilla version of TracIn. For BERT, more plots and a further discussion about TracIn are included in the Appendix (Section G.1). In the Appendix (Section G.2), we also illustrate the fading of influence for another NLP pre-trained model, T5. While we have illustrated the fading of influence for both BERT and ResNet, in the latter case the peak of correlation is quite low; we repeated the same experiments on the ViT (Section G.3), where we find a high correlation peak as in the case of BERT.

## 6 Using Influential Examples for Error Correction

[KL17] propose to correct model mis-predictions by first using influence scores to retrieve the examples most responsible for a given prediction, and, after correcting them, *retraining the model*. Computational considerations aside, a key conclusion from the theory in Section 3 and the empirical verification in Section 5.2 is that IF only predict influence over a limited number of training steps. In this section we demonstrate that IF can still be used to correct model mis-predictions by *taking a few fine-tuning steps on influential examples*.

---

[2]Further details in the Appendix.

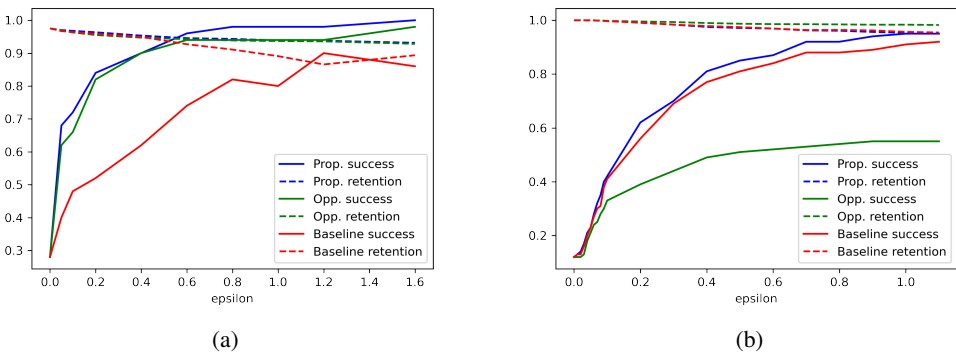

Figure 3: Success rate and retention for error correction. (a) BERT, (b) ResNet

Concretely, we propose to correct mis-predictions at a given test point $z$ by first identifying a batch of influential examples $B$ and then taking a few fine-tuning steps on the perturbed loss $\mathcal{L} = L + \varepsilon \cdot l_B$. We propose two methods: (1) *Proponents-correction*: we identify the set $B$ of top-k proponents for the current $x$ and *relabel* them to what should be the correct prediction on $x$; (2) *Opponents-tuning*: as opponents *oppose the current prediction*, we take $B$ to be the set of top-k opponents of $x$.

We investigate how well these error-correction techniques work on SST2 (BERT) and CIFAR10 (ResNet). As a baseline, we randomly sample a set $B$ of training points with the same label as the prediction on $x$ and then set their label equal to the correct one for $x$. We take a maximum of 50 fine-tuning steps and take the top-50 proponents or opponents to build $B$. The main metric we compute is the *success rate*, i.e. the ratio of mis-predictions successfully corrected within the limit of 50 steps. Additionally, we report *prediction retention* [DCAT21] on a fixed held-out set of 50 test examples, i.e. the ratio of examples predictions which have not changed after a correction – ideally, a correction does not cause too many changes in model predictions otherwise. We experiment with different values of $\varepsilon$, starting from no up-sampling and gradually increasing it to when influential examples account for slightly more than half of the batch.

From Figure 3 we see that Proponents-correction and Opponents-tuning strongly outperform the baseline in binary classification (SST2). For multi-class classification (CIFAR10) Opponents-tuning is not effective, as we verified that only 54% of the retrieved opponents have the desired label; Proponents-correction still outperforms the baseline increasing on average the success rate by 2% and reducing the number of steps to take by 6%. We conjecture that for ResNet the improvement over the baseline is less than for BERT because of the worse predictive power of IF (Figure 2 (b)). In Appendix H we include additional plots showing the number of steps to correct mis-predictions as a function of the up-sampling parameter $\varepsilon$.

The primary goal of the experiments in this section is to verify that tuning on influential examples results in a correction more reliably and faster than on other classes of examples. Since the SST2 and CIFAR10 datasets are largely clean, for the proponents method, the training example label is flipped to an incorrect one. In Appendix I we give examples from a noisy text classification dataset illustrating the scenario that the correction-with-proponents method is supposed to address.

## 7 Limitations

We derive the perturbed training trajectory (Theorem 2) and the divergence of trajectories (Theorem 3) for stochastic gradient descent and, while we sketch in the Appendix the modifications needed when using other optimizers, we do not pursue this topic in detail. In Section 6 we measure prediction retention after correcting mis-predictions. While this metric is intuitive and has been used previously (e.g., [DCAT21]), we do not distinguish between semantically similar and unrelated examples and thus do not check the consistency and generalization properties of the update [MBAB22], leaving a thorough study of the correction-retention tradeoff to future work.

# 8 Conclusions

IF have been regarded as a tool that promises to trace model behavior to the training data. Unfortunately, recent studies have found no empirical support for such a claim as IF fail to predict the LSOR effect. This finding gives rise to the question of what IF methods really predict and whether they could be useful for model debugging. In this work we clarified which questions IF can be expected to answer. We first identified problematic assumptions made by IF methods – a priori any of these assumptions could be a reason for the observed empirical failure of IF. Thus, for each assumption we studied if it is indeed problematic. While most have turned out to be addressable in one way or another, we demonstrated that the one about parameter divergence puts a severe limitation on IF. With a deeper analysis of this assumption, we revised what can be theoretically expected from IF: IF methods are time-bound, that is, they can at most predict what happens when fine-tuning on a perturbed loss for a limited amount of time. With that, a practical usage of IF for model debugging is still possible – we proposed and empirically validated a theoretically-grounded procedure to apply IF to correct model mis-predictions.

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

## A  Clarifications regarding the operator $\nabla$

We spell out in further detail the way we employ the $\nabla$ operator. For example, we say that $\nabla_{\varepsilon|0}\theta_\varepsilon$ is the $Q \times N$-dimensional Jacobian at $\varepsilon = 0$: concretely, $\nabla_{\varepsilon|\varepsilon_0} f$ means computing the Jacobian of $f$ wrt. $\varepsilon$ and evaluating it at the point $\varepsilon_0$; this Jacobian is a matrix whose $(i,j)$-entry is given by:

$$(\nabla_{\varepsilon|\varepsilon_0} f)_{i,j} = \frac{\partial f_j}{\partial \varepsilon_i}(\varepsilon_0). \tag{9}$$

This is easily extended to higher order derivatives denote by $\nabla^k$; specifically we use $\nabla^2_{(\varepsilon,\theta)|(0,\theta_0)}\mathcal{L}$ to denote the matrix whose $i,j$-entry is given by:

$$(\nabla^2_{(\varepsilon,\theta)|(0,\theta_0)}\mathcal{L})_{i,j} = \frac{\partial^2 \mathcal{L}}{\partial \varepsilon_i \partial \theta_j}(0,\theta_0); \tag{10}$$

as the matrix $H_{\theta_0}^{-1}$ acts on the space of the parameters $\theta$, the notation $H_{\theta_0}^{-1}\nabla^2_{(\varepsilon,\theta)|(0,\theta_0)}\mathcal{L}$ will denote a contraction on $j$:

$$(H_{\theta_0}^{-1}\nabla^2_{(\varepsilon,\theta)|(0,\theta_0)}\mathcal{L})_{i,k} = \sum_j (H_{\theta_0}^{-1})_{k,j} \frac{\partial^2 \mathcal{L}}{\partial \varepsilon_i \partial \theta_j}(0,\theta_0). \tag{11}$$

## B  Proof of Theorem 1

For convenience, we first recall the statement of Theorem 1.

**Theorem.** *Assume that $\nabla\mathcal{L}$ is $C^k$ ($k \geq 1$) and let $\theta_0 \in \mathbf{R}^N$; assume that the Hessian $H_{\theta_0} = \nabla^2_{\theta|\theta_0}L$ is non-singular; then there exist neighborhoods $U$ of $\theta_0$ and $V$ of $0 \in \mathbf{R}^Q$, and a $C^k$-function $\Theta : V \to U$ such that $\Theta(0) = \theta_0$ and $\Theta(\varepsilon) \in U$ is the unique solution in $U$ of the equation*

$$\nabla_\theta \mathcal{L}(\Theta(\varepsilon),\varepsilon) = \nabla_\theta \mathcal{L}(\theta_0,0). \tag{12}$$

*Moreover, the gradient of $\Theta$ at the origin is given by:*

$$\nabla_{\varepsilon|0}\Theta = -H_{\theta_0}^{-1}\nabla^2_{(\varepsilon,\theta)|(0,\theta_0)}\mathcal{L}. \tag{13}$$

*Proof.* Equation (12) gives us $N$ constraints on the $N + Q$ variables $(\theta,\varepsilon)$ and we want to solve them for the first $N$ variables $\theta$; to obtain the function $\Theta$ we invoke the Implicit Function Theorem, using that the Jacobian wrt. the variables we want to solve for is $H_{\theta_0}$ and is therefore non-singular. Finally (13) is obtained by taking the gradient of (12) wrt. $\varepsilon$ and setting $\varepsilon = 0$:

$$\nabla^2_{\theta|\theta_0}L \cdot \nabla_{\varepsilon|0}\Theta + \nabla^2_{(\varepsilon,\theta)|(0,\theta_0)}\mathcal{L} = 0.$$

$\square$

## C  Proof of Corollary 1

For convenience, we first recall the statement of Corollary 1.

**Corollary.** *Under the assumptions of Theorem 1:*

1. *If $\theta_0$ is a stationary point of $L$, then each $\Theta(\varepsilon)$ is a stationary point of the loss $\theta \mapsto \mathcal{L}(\theta,\varepsilon)$.*

2. *If $\theta_0$ is a local (strict) minimum of $L$, for $\varepsilon$ sufficiently small, $\Theta(\varepsilon)$ is a local (strict) minimum of the loss $\theta \mapsto \mathcal{L}(\theta,\varepsilon)$.*

3. *Let $Q = 1$ (hence epsilon is a scalar) with $\mathcal{L}(\theta,\varepsilon) = L(\theta) + \varepsilon l_x(\theta)$, where $l_x$ is the loss corresponding to a specific training point $x$. We then obtain the classical result [CS82] about the influence of down/up-sampling $x$ on the training parameters:*

$$\left.\frac{d\Theta}{d\varepsilon}\right|_{\varepsilon=0} = -(\nabla^2_\theta L(\theta_0))^{-1}\nabla_\theta l_x(\theta_0). \tag{14}$$

*Proof.* If $\theta_0$ is a stationary point of $L$, then $\nabla_\theta \mathcal{L}(\theta_0, 0) = 0$ in (3). This implies that $\nabla_\theta \mathcal{L}(\Theta(\varepsilon), \varepsilon) = 0$ in (3), which is exactly the statement that $\Theta(\varepsilon)$ is a stationary point of $\mathcal{L}$. If $\theta_0$ is a local (strict) minimum of $L$, it is not just a stationary point, but the Hessian at $\theta_0$ is also positive definite. By continuity, for sufficiently small $\varepsilon$, also the Hessian $\nabla^2_{\theta|\Theta(\varepsilon)}\mathcal{L}$ will be positive definite so that $\Theta(\varepsilon)$ will be a local (strict) minimum. Finally, (14) follows from applying (3) to the variation $\mathcal{L}(\theta, \varepsilon) = L(\theta) + \varepsilon l_x(\theta)$. $\qquad\square$

## D  Proof of Theorem 2

For convenience, we first recall the statement of Theorem 2.

**Theorem.** *Assume that the model is trained for $T$ time-steps with stochastic gradient descent. Denoting by $H_t$ the Hessian $\nabla^2_{\theta|\theta_{0,t}}L_t$, then the final parameters $\theta_{\varepsilon,T}$ satisfy:*

$$\nabla_{\varepsilon|0}\theta_{\varepsilon,T} = -\sum_{t=0}^{T-1}\eta_t\nabla^2_{(\varepsilon,\theta)|(0,\theta_{0,t})}\mathcal{L}_t - \sum_{t=0}^{T-1}\eta_t H_t \nabla_{\varepsilon|0}\theta_{\varepsilon,t}. \tag{15}$$

*Proof.* The parameters $\theta_{\varepsilon,t}$ obey a recurrence relation:

$$\theta_{\varepsilon,t} - \theta_{\varepsilon,t-1} = -\eta_{t-1}\nabla_\theta \mathcal{L}_{t-1}(\theta_{\varepsilon,t-1}, \varepsilon), \tag{16}$$

which can be solved to give

$$\theta_{\varepsilon,T} = -\sum_{t=0}^{T-1}\eta_t\nabla_\theta \mathcal{L}_t(\theta_{\varepsilon,t}, \varepsilon); \tag{17}$$

then (15) follows immediately by applying $\nabla_{\varepsilon|0}$, i.e. computing the Jacobian wrt. $\varepsilon$ at the origin. $\quad\square$

## E  Proof of Theorem 3

The first step in the proof of Theorem 3 is a discrete version of Gronwall's Lemma [Gro19].

**Lemma 1.** *Let $\{u_t\}$, $\{\alpha_t\}$ and $\{\beta_t\}$ be sequences such that $\beta_t \geq 0$ and*

$$u_t \leq \alpha_t + \sum_{s=0}^{t-1}\beta_s u_s. \tag{18}$$

*Note that we assume that* (18) *holds for $t \geq 1$ and we consider $u_0$ an initial condition. Then:*

$$u_T \leq \alpha_T + \beta_0\prod_{s=1}^{T-1}(1+\beta_s)u_0 + \sum_{s=1}^{T-1}\alpha_s\beta_s\prod_{k=s+1}^{T-1}(1+\beta_k). \tag{19}$$

*If $\{\alpha_t\}$ is non-decreasing in $t$ we then get:*

$$u_T \leq \beta_0\prod_{s=1}^{T-1}(1+\beta_s)u_0 + \alpha_T(1 + \sum_{s=1}^{T-1}\beta_s\prod_{k=s+1}^{T-1}(1+\beta_k)). \tag{20}$$

*Moreover, defining $u_t$ by setting* (18) *to be an equality, shows that* (19) *is sharp.*

*Proof.* We first observe that if we set $v_0 = u_0$ and build $v_t$ by declaring (18) to be an equality, then $v_t \geq u_t$ for any $t$. This is true by induction as:

$$u_{t+1} - \alpha_{t+1} \leq \sum_{s=0}^{t}\beta_s u_s \leq \sum_{s=0}^{t}\beta_s v_s = v_{t+1} - \alpha_{t+1}. \tag{21}$$

We thus focus on bounding $v_{t+1}$; we note that

$$(v_{t+1} - \alpha_{t+1}) - (v_t - \alpha_t) = \beta_t(v_t - \alpha_t) + \beta_t\alpha_t; \tag{22}$$

thus

$$v_{t+1} - \alpha_{t+1} = (1 + \beta_t)(v_t - \alpha_t) + \beta_t \alpha_t$$
$$= (1 + \beta_t)(1 + \beta_{t-1})(v_{t-1} - \alpha_{t-1}) + (1 + \beta_t)\beta_{t-1}\alpha_{t-1} + \beta_t \alpha_t; \tag{23}$$

then (19) follows by induction and letting $t + 1 = T$. Finally, if $\{\alpha_t\}$ is non-decreasing in $t$ we can simply replace $\alpha_s$ with $\alpha_T$ obtaining (20). The sharpness follows by considering the sequence $\{v_t\}$ we defined in the proof. $\qquad\square$

For convenience, we first recall the statement of Theorem 3.

**Theorem.** *In the setting of Theorem 2 assume that, for $t \le T$, $\theta_{\varepsilon,t}$ lies in a bounded region $R$ such that*

$$\sup_{\varepsilon,\theta \in R} \|\nabla \mathcal{L}_t(\theta, \varepsilon) - \nabla \mathcal{L}(\theta, 0)\| \le C\varepsilon \tag{24}$$

*and that for $\theta \in R$ each loss $\mathcal{L}_t(\theta, \varepsilon)$ and its gradient wrt. $\theta$ are $A$-Lipschitz in $\theta$. Then*

$$\|\theta_{\varepsilon,T} - \theta_{0,T}\| \le C\varepsilon \sum_{s<T} \eta_s (1 + \exp(2A \sum_{s<T} \eta_s)). \tag{25}$$

*Proof.* The idea is to apply Lemma 1 as we would in the case of a standard ODE. Let us compare the evolution of $\theta_{\varepsilon,t}$ and $\theta_{0,t}$:

$$\theta_{\varepsilon,t} = \theta_{\text{init}} - \sum_{s=0}^{t-1} \eta_s \nabla_\theta \mathcal{L}(\theta_{\varepsilon,s}, \varepsilon) \tag{26}$$

$$\theta_{0,t} = \theta_{\text{init}} - \sum_{s=0}^{t-1} \eta_s \nabla_\theta \mathcal{L}(\theta_{0,s}, 0); \tag{27}$$

which leads to

$$\|\theta_{\varepsilon,t} - \theta_{0,t}\| \le \sum_{s=0}^{t-1} \eta_s \|\nabla_\theta \mathcal{L}(\theta_{\varepsilon,s}, \varepsilon) - \nabla_\theta \mathcal{L}(\theta_{0,s}, 0)\|$$
$$\le C\varepsilon \sum_{s=0}^{t-1} \eta_s + \sum_{s=0}^{t-1} \eta_s A \|\theta_{\varepsilon,s} - \theta_{0,s}\|; \tag{28}$$

we now just need to rephrase this inequality in terms of Lemma 1:

$$\underbrace{\|\theta_{\varepsilon,t} - \theta_{0,t}\|}_{u_t} \le \underbrace{C\varepsilon \sum_{s=0}^{t-1} \eta_s}_{\alpha_t} + \sum_{s=0}^{t-1} \underbrace{\eta_s A}_{\beta_s} \underbrace{\|\theta_{\varepsilon,s} - \theta_{0,s}\|}_{u_s}; \tag{29}$$

we note that $u_0 = 0$ as both dynamics start at $\theta_{\text{init}}$ and that $\alpha_t$ is non-decreasing in $t$. We then have that

$$u_T \le \alpha_T (1 + \sum_{s=1}^{T-1} \beta_s \prod_{k=s+1}^{T-1} (1 + \beta_k))$$
$$\le \alpha_T (1 + \sum_{s=1}^{T-1} \beta_s \prod_{k=s+1}^{T-1} \exp(\beta_k))$$
$$\le \alpha_T (1 + \exp(\sum_{k=1}^{T-1} \beta_k) \sum_{s=1}^{T-1} \beta_s) \tag{30}$$
$$\le \alpha_T (1 + \exp(\sum_{k=1}^{T-1} \beta_k) \times \exp(\sum_{k=1}^{T-1} \beta_k))$$
$$\le \alpha_T (1 + \exp(\sum_{k=1}^{T-1} 2\beta_k)),$$

which is (8). $\qquad\square$

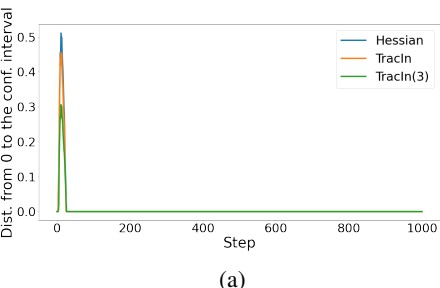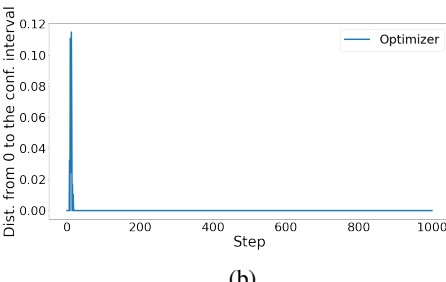

(a)                                    (b)

Figure 4: For BERT the predictive power of influence scores becomes 0 over time as the distance from 0 to the confidence interval of the Pearson's r becomes 0, i.e. the confidence interval contains 0.

## E.1   Sketching modifications needed in the case of optimizers

The proofs of Theorems 2 and 3 were given for SGD. The argument in the case of using an optimizer would be more involved. Here we sketch the modifications needed when dealing with optimizers. An optimizer is characterized by an *optimizer state*, $\sigma_{\varepsilon,t}$, which will also evolve in time. While for SGD we just considered the update rule for $\theta_{\varepsilon,t}$, in the case of optimizers one needs to study a joint system of update rules for the parameters and the optimizer state:

$$\theta_{\varepsilon,t} - \theta_{\varepsilon,t-1} = -\eta_{t-1}F(\nabla_\theta\mathcal{L}_{t-1}(\theta_{\varepsilon,t-1},\varepsilon),\sigma_{\varepsilon,t}) \tag{31}$$

$$\sigma_{\varepsilon,t} - \sigma_{\varepsilon,t-1} = -\rho_{t-1}G(\nabla_\theta\mathcal{L}_{t-1}(\theta_{\varepsilon,t-1},\varepsilon),\sigma_{\varepsilon,t-1}). \tag{32}$$

In the case of Theorem 2 one would then apply $\nabla_{\varepsilon|0}$ to the joint system to obtain the update rule. For Theorem 3 one should add additional continuity in $\varepsilon$ and Lipschitz conditions on $F$ and $G$; one should then check that these are indeed satisfied for common optimizers like Adam or Adafactor.

## F   Why do we use the Conjugate Residual method?

Regarding HIF, note that usually the Conjugate Gradient method is used [KL17] for computing inverse Hessian vector products. However, in our experiments the Conjugate Gradient method always yielded a time-series of Pearson Correlations $R(t)$ oscillating around 0, *thus without any predictive power*. The reason is that this method assumes the Hessian to be positive-definite, which is not the case for most Neural Networks. A simple fix to the problem is to use the Conjugate Residual method which does not require the Hessian to be positive definite. We recommend to use the Conjugate Residual when applying HIF in Deep Learning; *this might look like a minor technical point, but it can avoid reporting that HIF has no predictive power, when instead the issue lies in the numerical method used to compute inverse Hessian vector products.*

For more discussion on the Conjugate Residual method see its Wikipedia article.

## G   Further empirical results on Fading of Influence scores

### G.1   Fading of Influence for BERT

In the setting of Figure 2(a) we plot the distance from the confidence interval of the Pearson's r to 0 (Figure 4); once the distance becomes 0, the predictive power of influence scores has become null.

In Figure 5 we zoom in Figure 2 (a: BERT). The fading phenomenon was non affected by checkpoint selection: in Figure 6 we consider a later checkpoint and we see the same qualitative behavior. Moreover, in Figure 6 we also consider TracIn with 3 checkpoints selected using the advice in [PLKS20] – we do not see improvements and the peak is even slightly lower than for TracIn using one checkpoint. Thus, in our further experiments we have used TracIn with a single checkpoint.

In Figure 7 we zoomed in Figure 2 (b: ResNet). In this case the predictive power was never particularly high, e.g. for TracIn it quickly peaked at 0.3 but it takes more steps than in the case of BERT to reach 0. We conjecture that this is in part due to the slower divergence of parameters in

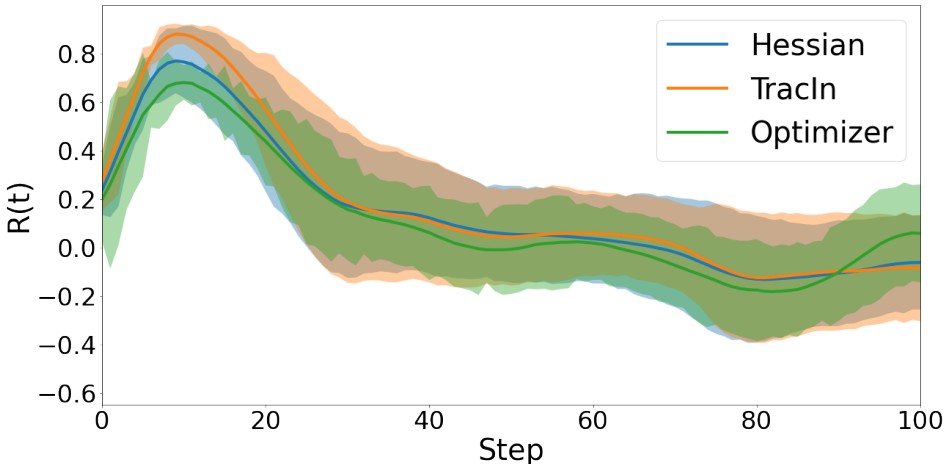

Figure 5: For BERT the predictive power of influence scores on the loss shifts degrades over time very quickly.

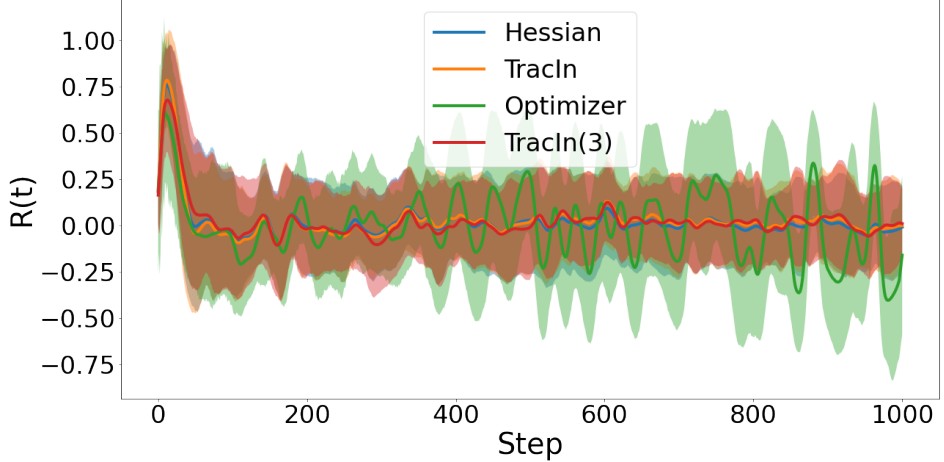

Figure 6: For BERT the predictive power of influence scores on the loss shifts degrades over time very quickly. Using multiple checkpoints did not improve performance of TracIn.

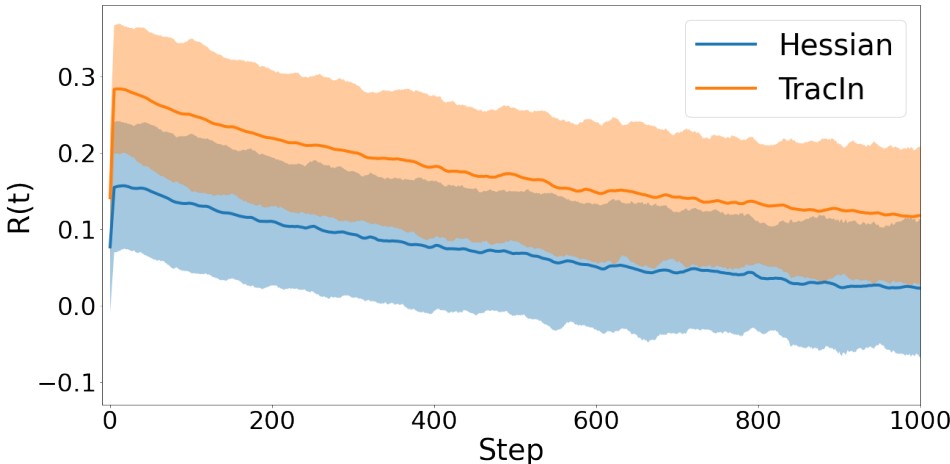

Figure 7: For ResNet the predictive power of influence scores faded more slowly and was never particularly high.

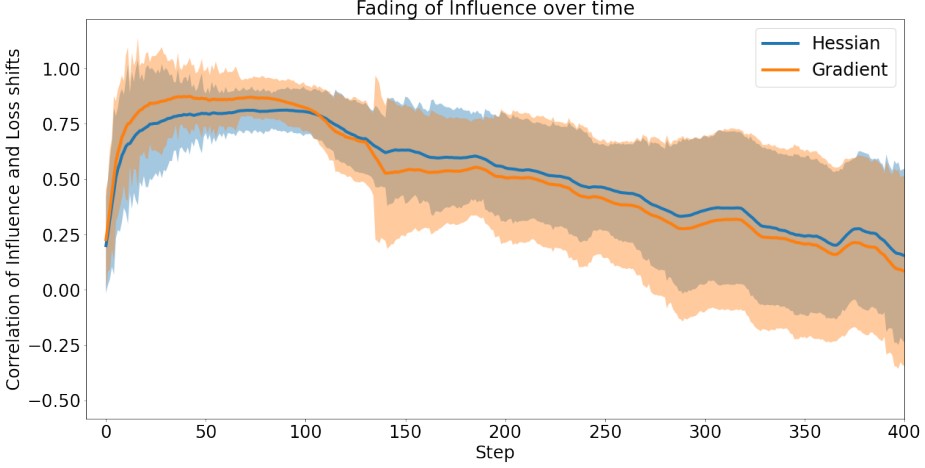

Figure 8: For BERT the predictive power of influence scores faded more slowly when using SGD. The "Gradient" method is TracIn.

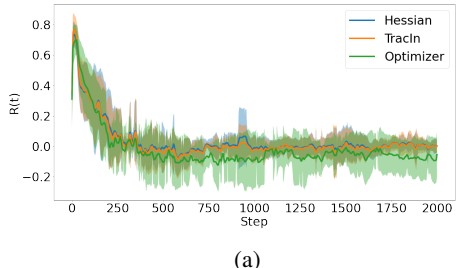 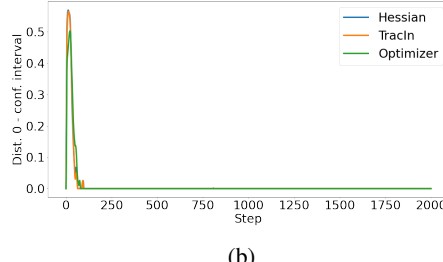

(a)                                     (b)

Figure 9: The predictive power of influence scores on the loss shifts degrades over time for T5. (a) Correlation with the ground truth, with the shaded area the corresponding $95\%$ confidence region. (b) Distance from 0 to the confidence interval, which measures when the confidence interval covers 0.

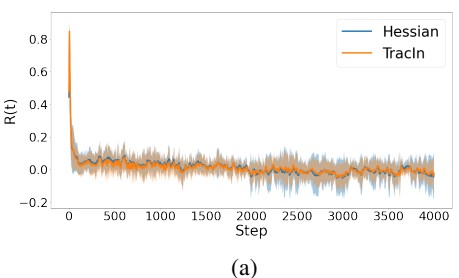 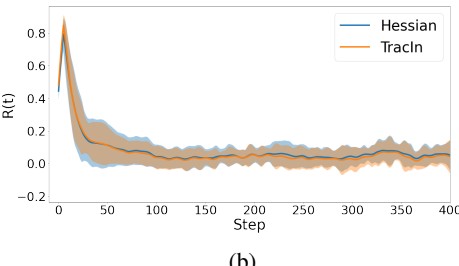

(a)                                     (b)

Figure 10: The predictive power of influence scores on the loss shifts degrades over time for the Vision Transformer (ViT). (a) Full range of re-training steps, (b) zoom-in in the peak and decay phase.

ResNet (compare the x-axis of (a) and (b) in Figure 1) and in part due to the use of SGD. In particular, we verified that a slower fading also takes place when fine-tuning BERT with SGD (Figure 8).

## G.2   Fading of Influence for T5

We consider another NLP pre- trained model fine-tuned on SST2, T5. In Figure 9 we illustrate the same effect of fading of influence for the T5 model.

## G.3   Fading of Influence for the ViT

In the case of ResNet, in our fading of influence experiments (Fig. 2 (b)) the peak is particularly low. On the same task, we trained a ViT with SGD and observe that the peak is higher, similarly to what happens with BERT, see Figure 10. We conjecture that this difference is due to the different model architecture. Note that in both cases, we observe the fading of influence as predicted by the theory.

# H   How many steps are needed to correct mis-predictions?

In Figure 11 we plot the average and median number of steps to correct a mis-prediction as a function of $\varepsilon$: for any fixed $\varepsilon$, the more effective a correction method is, the fewer steps are needed to correct the prediction. For BERT we see that both Proponent-correction and Opponent-tuning result in a large decrease in the number of steps to take, compared with the Baseline. For ResNet, we do not plot Opponent-tuning as it performed poorly on success-rate. For ResNet the gains of Proponent-correction are smaller but consistent as $\varepsilon$ varies.

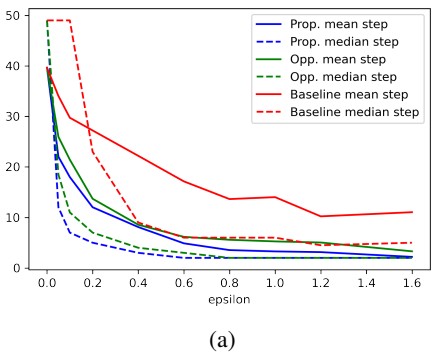
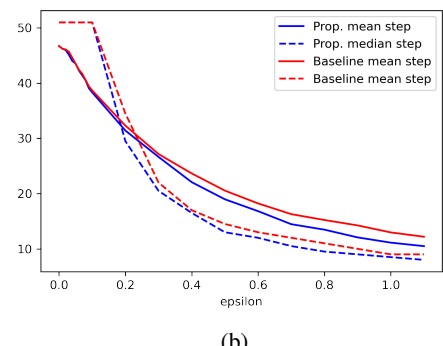

|                                | (a) | (b) |
| --- | --- | --- |

Figure 11: Average and median steps to correct mis-predictions. (a) BERT, (b) ResNet

Table 1: Pairs of misclassified test examples with their mislabeled training proponents

|   | Test example | Top proponent |
| --- | --- | --- |
| | **False positives** | |
| 1 | Agreed. This is dumb. | Yeah i no it sucks. |
| 2 | == Iowa-class battleships == Stupid me... I didn't even look... sorry about that... ( | By the way, sorry about that stupid edit. |
| 3 | It's so silly of me to ask this question but I'm still not getting what you're trying to say. Sorry for being such an idiot! ;) | By the way, sorry about that stupid edit. |
| 4 | hahaha this whale is huge and ugly and funky doodle...lol... | A cheeseburger for you! Fatty |
| 5 | Cool you can change the content!i love sex!! | Sex in space Sweet. Thanks for the education! ) |
| 6 | ===== A lot of poeple got killed that day ===== | Yes they were Hitler Stated that Slavs were lesser races and should be salves or killed |
| 7 | == A question regarding Rodriguez's sexual orientation == Would the statement, "I've gone both ways. I do as I please. I am too fucking curious to sit here and not try when I can. Men are intriguing. So are chicks." be enough to label her as bisexual or LGBT? | Ravenkills97 I didn't change the genres on In This Moment, all i did was put commas in between the genres. U gave me a final warning for no reason. All because I put commas in between something, like it fucking erases the genre. |
| | **False negatives** | |
| 1 | your mama is a man | grow up Get a life |
| 2 | (...) It's nice to see you finally admit it with your edit summary of, "what a terrible idiot I am". (...) | (...) you're a nerd living with your mom and you have no friends. You see? (...) |
| 3 | U r worse than obamas presidency | Are you known as a sock puppet? |
| 4 | I don't want to say sorry ,so get a life before i ruin yours buthead | Then grow up and stop spamming on my page. I hear your mommy calling. Run along. |
| 5 | (...) it's hard for me to not take the kind of stupidity evident from your edits personally (...) | (...) thats 2 against 1 buddy you lose |
| 6 | Your spelling seems to be at about the 3rd grade level, I hope you're not having sex. | Okay then. I'll just have to agree to disagree. I won't read your article and I won't give you fellatio. Now get off my page. |

Table 2: Mean influence with variance and metrics for the experiments with the example in Table 1

| proponents | mean influence | mean step | median step | retention |
|---|---|---|---|---|
| True | 0.91 ($\pm$0.03) | 3.0 | 2.5 | 98.1% |
| Permuted | 0.50 ($\pm$0.21) | 5.1 | 3.5 | 96.3% |

Table 3: Training hyper-parameters for ResNet on CIFAR10

| Hyper-parameter | value |
|---|---|
| Batch-size | 128 |
| Epochs | 200 |
| Learning rate | $10^{-1}$ |
| Learning rate scheduler | cosine decay |
| Optimizer | SGD |
| $l_2$-regularization | $10^{-4}$ |

# I   Selecting mislabeled proponents for correction

For the experiments in Section 6 we use comparatively clean datasets, so the proponents we retrieve and correct are not mislabeled. In this section we use the notoriously noisy Wikipedia Toxicity Subtypes dataset [WTD17] where for many misclassified test examples we can find mislabeled proponents used during training. Table 1 presents false positives and negatives (that is, test examples incorrectly predicted to be toxic, resp. non-toxic) with a high-scoring proponent which we consider mislabeled. It is worth noting that some examples lack context necessary to confidently verify whether they are toxic or not. In total, we collected 14 such pairs, seven for mispredictions of each kind.[3]

Table 2 presents the mean and the median number of steps needed to be taken on the corrected proponents to achieve a change in prediction, as well as retention once there is a change. As a point of comparison, we permute the proponents withing the false positive and false negative sets: this is a more challenging baseline than selecting examples for correction randomly. Note that the influence scores for the presented examples range from $-0.90$ to $0.92$, so some of the permuted proponents still have some quite influence on the test examples they get assigned to. Still, the results confirm that correcting mislabeled top-scoring proponents is the fastest way of achieving a change in model predictions, with as few as three steps required on average and with the smallest damage to predictions on other examples (measured with retention).

# J   Hyper-parameters

## J.1   Training

ResNet was trained on a single V100 with the hyper-parameters in Table 3; training data was augmented using `torchvision` using the transformations `transforms.RandomCrop(32, padding=4)`, and `transforms.RandomHorizontalFlip()`.

---

[3]We omitted the most upsetting example from the false negatives.

Table 4: Training hyper-parameters for BERT on SST2

| Hyper-parameter | value |
|---|---|
| Batch-size | 128 |
| Steps | 35000 |
| Learning rate | $10^{-5}$ |
| Learning rate scheduler | None |
| Optimizer | Adam |
| $l_2$-regularization | $5 \times 10^{-6}$ |

Table 5: Training hyper-parameters for T5 on SST2

| Hyper-parameter | value |
|---|---|
| Batch-size | 16 |
| Maximal sequence Length | 128 |
| Epochs | 4 |
| Learning rate | $10^{-4}$ |
| Learning rate scheduler | None |
| Optimizer | Adam |

Table 6: Training hyper-parameters for ViT on CIFAR10

| Hyper-parameter | value |
|---|---|
| Batch-size | 128 |
| Epochs | 300 |
| Peak learning rate | $10^{-3}$ |
| Learning rate scheduler | 1 epoch warm-up and then cosine decay |
| Optimizer | SGD |

The ViT was trained on a single V100 with the hyper-parameters in Table 6; we used the same data augmentation as in the case of ResNet.

BERT was trained on 8 TPUv3 cores using the hyper-parameters in Table 4. The best checkpoint was selected on validation-set accuracy evaluated every 500 steps; it corresponded to 6000 steps of training.

T5 was trained on a GPU V100 using the hyper-parameters in Table 5. We selected a checkpoint based on the validation set accuracy at epoch 4, where the accuracy was 92.9%. Note that T5 is a generative model, so at prediction time the sequence to classify is encoded with the encoder and then the decoder is used to generate a sequence that represents the desired class label. When computing influence functions, one uses the cross-entropy loss for sequences.

### J.2 Correction Experiments

For the correction experiments we used ABIF [SZTS22] because of their computational efficiency as IF scores need to be computed against the whole training set; we used 32 projectors obtained with 64 Arnoldi iterations. The Arnoldi iteration can take up to 2 hours; scoring the data takes a few minutes. We used a learning rate of $10^{-3}$ and for BERT we loaded the state of the Adam optimizer from the selected checkpoint.

For BERT we selected the checkpoint at 6000 steps which was the best on validation performance. For ResNet we selected the checkpoint after 10 epochs at which about 150 test points are incorrectly classified.

## K    Illustrating TracIn's modeling error

In Theorem 2 we have derived the correct formula for the Jacobian $\nabla_{\varepsilon|0}\theta_{\varepsilon,T}$ and we expect TracIn (Eq. 2) to introduce a modeling error in calculating $\nabla_{\varepsilon|0}\theta_{\varepsilon,T}$. We can compute the difference between the ground truth $\theta_{\varepsilon,T}$ and the first order expansion $\theta_{0,T} + \varepsilon^T \nabla_{\varepsilon|0}\theta_{\varepsilon,T}$ when the Jacobian is computed with TracIn (Eq. 2) or the corrected formula (Eq. 6). We can then compute the relative excess error introduced by TracIn for different values of $\varepsilon$. In Figure 12 we observe that except for the extremely small perturbation $\varepsilon = 10^{-4}$, TracIn does introduce an additional relative error in modeling the change of parameters when retraining BERT with SGD. The setup of this experiment is the same as those of our parameter divergence experiments.

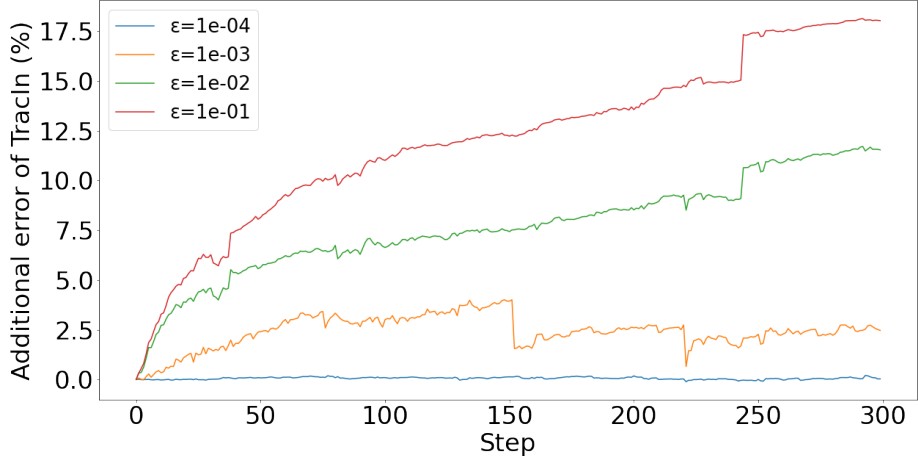

Figure 12: Illustrating the additional modeling error introduced by TracIn for different values of the perturbation $\epsilon$. The error is the ratio between the error of TracIn (Eq. 2) and the error of the exact formula (Eq. 6) when retraining BERT on a perturbed loss with SGD.

## L    Broader Impact Statement

We believe that our work can benefit model developers who want to debug and correct mis-predictions made by models. Our suggested error-correction procedure is much less compute intensive than previous ones using IF as it does not require model retraining. However, this debugging procedure could introduce new errors in the systems, for example if on a specific problem IF are not effective at predicting loss-shifts or if examples retrieved by IF lead to over-fitting against spurious artifacts present in such examples. Since this could have a negative impact on the users of such systems, mitigation strategies should be put in place regarding the trade-offs between the success rate of fixing specific mis-predictions and the overall retention of the system performance.

