# OpenReview forum: "Theoretical and Practical Perspectives on what Influence Functions Do"
_NeurIPS.cc/2023/Conference — NeurIPS 2023 spotlight_

### Official Review · Reviewer_qity · 2023-07-05

**Soundness:** 4 excellent
**Presentation:** 4 excellent
**Contribution:** 3 good
**Rating:** 7
**Confidence:** 4

**Summary:**

This paper reexamines the assumptions used in the deduction of influence function (IF) methods in order to explain the failure of IF in predicting leave-some-out-retrain performance. The authors find that all five assumptions used in the previous deduction will be violated to different degrees in practice and propose a combination of HIF and Arnoldi-based methods to overcome these violations. They find that four of the assumptions can actually be bypassed or fixed, however, the remaining challenge named *parameter divergence* seems inherent. They show that the predictive ability of IF will gradually fade over training steps due to this phenomenon and show that accounting for this effect, it is better to interpret IFs as proxies for the effect of a few fine-tuning steps.

**Strengths:**

* **Originality.** This paper shows an in-depth analysis of the assumptions on which IFs are based. Their analysis combines theoretical thinking well with experimental observations.
* **Clarity.** The results are presented in a clean way and are a pleasure to read.
* **Significance.** Although the reviewer is not an expert in the field of influence functions, the question this paper targets seems important and the paper sheds light on the real obstacles to solving this problem, as well as proposes possible ways to solve it.

**Weaknesses:**

The reviewer thinks the paper may improve in the following aspects.

* The authors argue that (1) using Arnoldi-based methods helps boost the accuracy of HIF, and (2) accounting for training trajectories can improve the estimation for IFs. The theoretical deduction of these propositions is very reasonable but it would be better to include ablation experiments or cite relevant literature to showcase the phenomenons.

* The authors observe that IF methods perform poorly on Resnet but do not have any explanation for this phenomenon. It would be beneficial if the authors can investigate the reason behind the difference of IF methods performance on NLP and CV tasks.

**Questions:**

The reviewer is interested in the following questions.
 * Why does the correlation between IFs and performances seem to increase before decreasing?
 * How would the authors suggest modifying the current IF estimation methods based on their observations?
 * What is the author's explanation of the rapidly increasing phase 1 in figure 1?

**Limitations:**

The authors have adequately addressed the limitations and the reviewer has not noticed any potential negative societal impact.

---

> ### Author Rebuttal · Authors · 2023-08-08
>
> **Weaknesses bullet point #1**:  For (1) we can point the reader to two relevant references. First, Schioppa et al. who compare  ABIF with exact Hessian on retrieving mislabeled examples (Fig. 2). Second, Fisher et al. (``Influence Diagnostics under self-concordance’’; Fig. 3 and the discussion in Sec 5.2) where they find that  ABIF is the best solver on the QA task and is comparable to the other solvers on the text-completion task. (2) is a natural research direction; at the moment Theorem 2 gives an exact formula but making it practical requires further ideas. We will include an ablation showing the extra error introduced by TracIn vs the exact formula in Theorem 2 in tracking parameter changes, if the paper gets accepted.
>
> **Weankesses bullet point #2**: We focused on the limitations of the methods, in particular the fading of predictive power. We conjecture that the low correlation is due to the architecture. Indeed, using a ViT for the CV task leads to a peak correlation similar to that of the NLP task. We will include the ViT result in the final version, if accepted. We leave an explanation of the phenomenon, possibly because of the different loss landscape geometry between the ResNet and ViT, as a question for future work.
>
> **Question regarding correlation increasing and then decreasing**:  When changing the loss to the perturbed one there is a phase of adjustment in the network. Using a dynamical analogy, the momentum the system had built on the old loss needs to change for the new loss and this introduces a lag to reach the peak performance.
>
> **Question regarding how authors would suggest modifying IF estimation**:  To improve estimation quality: start from formula (6) which traces the dynamics exactly; however (6) is not practical so techniques need to be developed to make it efficient. To manage expectations and evaluate methods: proceed as in the fading of influence experiments identifying the time window in which correlation is good enough.
>
> **Question about the rapidly increasing phase in Figure 1**: This is honestly a hard question. Given the network's complexity, we think that the $A$ in the (theoretical) upper bound in 4.4 is quite large; for the $A$ that comes (empirically) in the lower bound we think that there are two regimes: 1) early regime were the the network tries to adapt quickly to the perturbed loss; 2) asymptotic regime in which a lower value of $A$ is needed to adjust on the perturbed loss. As BERT (a) uses the Adam optimizer, in that case it also seems the phase is very rapid because of the quickness of the optimizer.

---

> > ### Comment · Reviewer_qity · 2023-08-11
> > **Response to Rebuttal**
> >
> > Extended results on ViT will definitely improve the paper.
> >
> > The reviewer has read the response and decided to keep the score.

---

### Official Review · Reviewer_SKsY · 2023-07-06

**Soundness:** 4 excellent
**Presentation:** 4 excellent
**Contribution:** 4 excellent
**Rating:** 8
**Confidence:** 4

**Summary:**

Influence functions allow one to measure the effect of up-weighting a training sample on the test loss (or functions of the model parameters) of a test example. This paper studies several of the assumptions needed for hessian-based influence functions to produce valid leave-out-one estimates of the effect on a test sample's loss. This assumptions include: convexity, stability of the hessian, and additivity of training trajectory. The paper shows that some of these assumptions are not as problematic, in practice, as previously assumed. They show that several of these assumptions can indeed be satisfied with changes to the formulation; for example, if the hessian becomes degenerate, one can approximate the hessian via arnoldi iteration. It turns out that the most problematic assumption results from a divergence between the initial set of parameters, and another obtained by re-training or even fine-tuning for longer time steps. The paper then bounds this parameter divergence using a discrete version of gronwall's lemma, and gives a set of takeaways for how to evaluate influence functions: the takeaway from this work is that influence scores only predict effect of a training sample over a small number of training steps. In the final portion of the paper, they evaluate how to use influence functions to correct errors.

**Strengths:**

Overall, I really enjoyed reading this paper, and found the breakdown of the assumptions really comprehensive.

**Quality/Clarity**\
I found the paper to be of very high quality. Each assumption is stated clearly and cleanly discussed. For example, I found the resolution of assumption #1 to be interesting. Here the authors show that for hessian-based influence functions one does not need to assume strict convexity but we can relax it to assume that the final gradient steps do not change the hessian by too much. While this might seem like a simple change, it helps provide clarity to this literature. The bigger takeway from this work is that influence scores only predict effect of a training sample over a small number of training steps. This finding is important since it requires rethinking how these approaches are currently used. Overall, this paper sets out key assumptions of influence functions, discusses how to remedy violations to these assumptions, and presents experiments to corroborate the theory presented.

**Significance**\
One important takeaway is that, for deep models, it now makes sense that influence scores obtained via retraining have poor correlation with leave-one-out retraining scores. Secondly, another important takeaway is that one can correct errors simply be reweighting or fine-tuning on opponents or proponents. The findings from the paper should be important for practitioners and others using influence functions as part of their debugging toolbox.

**Originality**\
This paper mostly considers a question that was also studied by Bae et. al. (Neurips 2022), but does so in a different way and using mostly new techniques. I found the use of the discrete version of gronwall's lemma to be quite nice. I am not sure if this is new, but taken together, this work advances the understanding of influence functions quite substantially.


**Weaknesses:**

The main weakness I have with this work are minor and listed below:

**Relation with Bae et. al. from Neurips 2022**: The main related work to this one is this paper. However, this work does not do enough to sufficiently contrast their analyses with theirs. I am mostly familiar with this previous work, and understand that they address exactly the same question as well, but arrive at a different conclusion. Their finding is that hessian-based influence functions approximate what they term the "proximal bregman response function", and that the ability of this quantity to match a sample loo estimate is mostly affected by the linearization error (i.e., due to the taylor expansion), and 2) hessian approximation. This work seems to not agree with theirs, i.e., you suggest that hessian approximation be handled with arnoldi, and that the key discrepancy comes from parameter divergence. It would be great if the authors could spend more time in the draft to contrast this work against that one. I think this is an important issue to address in the paper.

**Questions:**

Some clarification questions:

- For error correction, if I have k examples that I want to correct, then am I correcting each example independently or the taking gradient steps on all k examples at the same time. If it is independent, then isn't this too onerous?
- One use of these influence scores is mislabelled training samples, how does assumption 5 affect that? When in training should the self-influence score be a useful metric for mislabeled label correction?
- This is minor question, going from $\theta_\epsilon \approx \theta_0 + \epsilon^\top\nabla \theta_\epsilon$ to equation 1, where did the second $\nabla_\theta$ come from? Is this the chain rule? I am familiar with other ways to derive influence functions.

**Limitations:**

The authors discuss limitations.

---

> ### Author Rebuttal · Authors · 2023-08-08
>
> **Comparison to Bae et. al**:  Thank you for the suggestion, we will expand the discussion of Bae et al. Some comparison points:
>
> * Bae: Only Hessian-based influence functions; Us: we discuss Hessian and Gradient-based influence functions covering e.g., also TracIn.
> * Bae: convexity assumption is crucial (they add a regularization term); Us: we can drop convexity. Note that dropping convexity assumption implies one needs to use another solver (Conjugate Residual instead of Conjugate gradient).
> Under the convexity assumption there is agreement with their findings in the case of Hessian-based influence functions. However, advocates of the TracIn method might argue that they trace the training dynamics correctly while the Hessian-based IF does not and so their method would not be affected by the linearization error. However, we show that parameter divergence puts a limitation on TracIn too.
>
> **Question about $k$-examples for error correction**: We looked at $k=1$ as it matches the case of leave-on-out eval. But the method can be applied also on k examples by retrieving proponents of each test example in order to build $B$. In Figure 8 (Appendix) the median steps for correction are not so big, so for $k$ not too large doing one example at a time might not be that onerous. That being said, we agree that a proper understanding of the utility and limitations of IF for error correction deserves a deeper analysis. Given that the current submission focuses on the theoretical aspects, we leave such an analysis for future work.
>
> **Question regarding usage of self-influence scores**: To find outliers with self-influence, assumption 5 might not be so limiting. What we have seen in practice is that outliers tend to have big gradients so they get high self-influence scores. If the gradients stay big during training because of mislabeling then it does not really matter at which checkpoint they get measured. 5 affects the interaction between a test point and its proponents and opponents which is a finer quantity to measure.
>
> **Question regarding the second $\nabla$**: One $\nabla$ is with respect to the parameters $\theta$; the second $\nabla$ is with respect to the variation parameter $\varepsilon$; to get influence we need to map the loss to gradient space (hence $\nabla_\theta$) and then take the derivative with respect to $\varepsilon$ ($\nabla_\varepsilon$) which we denoted as a $\nabla^2_{\varepsilon,\theta}$. We will clarify the notation in the final version if the paper is accepted..

---

> > ### Comment · Reviewer_SKsY · 2023-08-15
> > **Response to rebuttal**
> >
> > Thanks to the authors for responding to my questions. The response here w.r.t. Bae et. al. is important, and belongs in the paper. To me that is the most relevant related work to this one. I maintain my rating.

---

### Official Review · Reviewer_RiS8 · 2023-07-06

**Soundness:** 3 good
**Presentation:** 3 good
**Contribution:** 3 good
**Rating:** 7
**Confidence:** 3

**Summary:**

This paper studies the limitations of Influence Functions (IF) based on assumptions about convexity, numerical stability, training trajectory and parameter divergence. The authors discuss each of those assumptions in detail and propose ideas to address the shortcoming of those limitations. The authors describe solutions for 4 assumptions made in the paper and discuss the limitations of the fifth assumption, parameter divergence, which is harder to address.
Apart from that the authors also illustrate the theory of parameter divergence on the example of NLP and vision classifiers. They confirm theoretical findings of fading influence over time when retraining the models on vision and test classifiers.
The authors also show how to perform error correction based on proponent-correction and opponent-tuning using a few fine tuning steps. They show that their approach outperforms baseline re-training procedures.


**Strengths:**

+ The paper is well written, especially the introduction, problem statement and the assumptions are well framed.
+ The related work is well cited and based on the studies in the recent papers, this paper proposes to address the limitations highlighted in the literature such as the LSOR problem for influential examples.
+ The paper studies 5 important assumptions made by IF and proposes solutions that help to alleviate the limitations posed by those assumptions. It also discusses and proposes a solution for the temporal dependency which is not accounted for in the original TracIn paper.



**Weaknesses:**

+ The main contributions of the paper seem to be around studying 5 different assumptions and proposing solutions for them. The main contribution is for assumptions 3 - 5 where the authors show that IF can predict \theta_{\epsilion, t} only for a limited period of time and propose incorporating temporal term into the tracin's formulation. These are important findings, however, it is not very clear whether it is a large enough contribution and whether it is completely novel since similar limitations of Tracin were also discussed in the Kelvin Guu, et.al.
+ It looks like all the proofs corresponding to theorems mentioned in the main paper are in the appendix. Since it is not required for reviewers to read appendices it might be good to consider moving some of the important proofs or parts of them into the main paper.
+ Section 4.2 might be difficult to follow for someone if they do not already know about Arnoldi. There is a brief description that Arnolodi approximates P1 using subspaces spanned by eigenvectors corresponding to top-k eigenvalues but I think that overall intuition of Arnoldi is not clear if someone is not familiar with that work.
+ Overall I think that Section 4.2 might be a bit difficult to follow (e.g. how is Theorem 1 applied to restricted variation).
+ It feels that it is somewhat difficult to follow how we arrive from Eq 7 to Eq 8. Perhaps adding more explanation in section 4.4 would be good.

**Minor comments**

+ The definition of C^k function on line 179 is a bit unclear. C gets used again in line 234 (C\epsilion) which makes it a bit confusing in terms of the readability of the paper.
+ Figure 3: X-axis is not annotated


**Questions:**

+ Section 6: Proponents-correction: when the set B is identified and relabeled, is it done for truly mislabeled examples ? How were those mislabeled examples sourced ?  For experiments with SST and ResNet how were set Bs curated ?
+ Section 6: Which formulation of IF (TracIn or Influence Functions) was used in the experiments ?
+ Section 5.2: Is R(t) pearson correlation between losses before and after re-training averaged for the same batches? Does the `Step` indicate different checkpoint on Figure 2 ?


**Limitations:**

The authors discuss several limitations of their work and propose ideas to address those limitations in the future.

---

> ### Author Rebuttal · Authors · 2023-08-08
>
> **Comparison to SimInfluence**:  A key difference is that, while Guu et al. (Simfluence) discuss the limitation of additivity in TracIn from a modeling perspective, we point at the limitation from its theoretical derivation. Our *first step* is showing that the original derivation is incomplete: starting from the same modeling assumptions of TracIn we derive the correct result that takes the time order into account. This is in contrast with Guu et al. who do not challenge the original derivation of TracIn (e.g. the role of the time order) but propose to extend TracIn with multiplicative terms. The *second step* is then showing that even if one were to use the corrected formula, one would have to face the issue of parameter divergence – something that Guu et al. do not discuss. We show that this issue is common to all the methods that are formulated with a first order expansion in $\varepsilon$, e.g. also the Influence Functions of Cook and Weisberg or the more recent ABIF.  In sum, while it is true that both Guu et al. and we point out the additivity limitation, this is probably the only similarity: everything else, including the motivation, the theoretical framing and the experiments is very different.
>
> **Question regarding Proponents correction and mislabeled examples**: We use comparatively clean datasets, so the proponents are not mislabeled. However, our primary goal in the correction experiments is to verify that intervening on influential examples results in a faster correction than on other classes of examples. Also, please note that this simplification applies to proponents (whose likely correct label is changed) but not to opponents (which are not changed). Finally, we also did an experiment with a notoriously noisy Wikipedia Toxicity Subtypes dataset where we indeed identified a mislabeled proponent for every test error. We will include these results in the appendix.
>
> **Question regarding IF method used in Section 6**: ABIF (Arnoldi-based influence functions) for scalability reasons. So it is basically a scalable and numerically stable version of Influence Functions.
>
> **Question regarding Section 5.2 and $R(t)$**: For each pair $(i, j)$ ($i$ in test and $j$ in train) we do re-training to compute the loss-shifts at each time step (i.e. checkpoint, index $t$) obtaining a tensor $A_{i,j,t}$; for each $i$, $j$ and $t$.
> Influence functions at the starting checkpoint give a score $s_{i,j}$; so we compute the Pearson correlation on the $(i,j)$ pairs between $A_{i,j,t}$ and $s_{i,j}$ to obtain R_t. For each experiment run the pairs $(i,j)$ are always fixed at the beginning (as we need to do a re-training run for each $j$); however across different runs we use different $(i,j)$ to build the confidence estimates for $R_t$.
> Step indicates thus a different checkpoint, but the checkpoints correspond to each step; since the sequence of models fits on the CPU RAM, one just needs to offload the parameters at each step to the CPU.

---

> > ### Comment · Reviewer_RiS8 · 2023-08-14
> > **Rebuttal Response**
> >
> > Thank you authors for the clarification and detailed responses to my questions. I've increased the rating by 1 point.

---

### Official Review · Reviewer_7Fho · 2023-07-21

**Soundness:** 3 good
**Presentation:** 3 good
**Contribution:** 3 good
**Rating:** 7
**Confidence:** 2

**Summary:**

This paper verifies assumptions of Influence Functions that show discrepancies in theory and empirical results, convexity, numeric stability, training trajectory, and parameter divergence. Although the other assumptions are addressable, parameter divergence is not. Hence, the paper proposed a solution to it and also showed empirical results.

**Strengths:**

-	Based on their observation of the issue, they proposed a simple solution to it.
-	To the reviewer, this paper reads as an instructional paper, which can contribute to the community.
-	It verified the applicability of the IF assumption one by one on convexity, numeric stability, training trajectory, and parameter divergence.


**Weaknesses:**

-	Aside from its technical contribution, there’s room where its writing can be improved. In particular, the introduction can be better structured, and there are multiple repetitive places that can be concise.
-	In Fig 2 (1) BERT case, although the authors claimed that the predictive power degrades monotonically over time, the “fading” effect is actually not clear. There’s only one time of peak at an early step, and there's no decreasing trend in the later steps. There is no fading effect either in the Appendix (Fig. 5.)



**Questions:**

Do the authors have other models to show the fading effect? As BERT’s case is unclear, only one empirical case of ResNet shows it.

**Limitations:**

Yes.

---

> ### Author Rebuttal · Authors · 2023-08-08
>
> **Reply to Weaknesses bullet point 2 & Questions**
>
> *Regarding the fading effect for BERT results*: Multiple runs (n=9) were used to build confidence intervals; the goal is to show that eventually 0 falls inside the confidence interval. For the methods Hessian, TracIn, TracIn(3) we observe that 0 falls in the confidence interval.
> For Optimizer this was not actually the case and there was sometimes a small distance of up to ~0.1 to ~0.2 from the confidence interval; however the mean of the time series is oscillating quickly up and down around 0 so we conjectured that this effect was due to the high variance of computing influence scores with the Optimizer adjustment. We have done additional runs (for a total of n=25) and we find that then also for Optimizer 0 is eventually covered by the confidence interval. Please see Figure 1 in the rebuttal
> plots .pdf. In Figure 2(a) we plot the confidence interval for Optimizer with those 25 runs and it covers 0. As our proof is formulated for SGD, we additionally re-did the BERT experiments when using SGD instead of Adam: see Figure 2(b); the fading effect there is particularly clear.
> We hope that these plots illustrate the effect convincingly for BERT and we will include them in the final manuscript. We will add another NLP pretrained model  and the ViT for the Vision Task for the camera-ready version if accepted.

---

> > ### Comment · Reviewer_7Fho · 2023-08-16
> >
> > Thanks for the response. I maintain my rating.

---

### Author Rebuttal · Authors · 2023-08-08

 We thank the reviewers for all the comments and suggestions and the positive feedback! Concerning readability (e.g., moving some proofs from the Appendix to the main body, explaining Arnoldi, explaining $C^k$-differentiability, Theorem 1 and Eq. (7-8) in more detail and improving the plot presentation), we will incorporate the suggestions in the final version, if the paper is accepted.

---

### Decision · Program_Chairs · 2023-09-21

**Decision:**

Accept (spotlight)

**Comment:**

Influence functions are used to measure the influence of a training sample on the decision on a test sample. The paper studies in a didactic way the assumptions considered for producing via influence functions,  approximations of leave one out estimates of the effect of removing a training sample. The study covers hessian based IF and gradient based IF such as traceIn.

These assumption are convexity, stability of the hessian, and additivity of training trajectory and parameter divergence. While most these assumptions can be problematic , they can be adressed , but  the study hints that the parameter divergence is the most problematic as the influence fades over iterations and can be only determined for a small number of  training iterations. Authors showcase then how influential points can be used in fine-tuning to improve the accuracy of models in NLP and computer vision tasks.

All reviewers agreed on the relevance of the work and clarified few minor issues in the paper, please incorporate the reviewers feedback and explicit the comparison to Bae et al in the related work and discuss it, Accept